# Bioaffinity Nanoprobes for Foodborne Pathogen Sensing

**DOI:** 10.3390/mi14061122

**Published:** 2023-05-26

**Authors:** Tracy Ann Bruce-Tagoe, Michael K. Danquah

**Affiliations:** Department of Chemical Engineering, University of Tennessee, Chattanooga 615 McCallie Ave, Chattanooga, TN 37403, USA; pvd756@mocs.utc.edu

**Keywords:** aptamer, nanoprobe, biosensor, bioaffinity, food safety

## Abstract

Bioaffinity nanoprobes are a type of biosensor that utilize the specific binding properties of biological molecules, such as antibodies, enzymes, and nucleic acids, for the detection of foodborne pathogens. These probes serve as nanosensors and can provide highly specific and sensitive detection of pathogens in food samples, making them an attractive option for food safety testing. The advantages of bioaffinity nanoprobes include their ability to detect low levels of pathogens, rapid analysis time, and cost-effectiveness. However, limitations include the need for specialized equipment and the potential for cross-reactivity with other biological molecules. Current research efforts focus on optimizing the performance of bioaffinity probes and expanding their application in the food industry. This article discusses relevant analytical methods, such as surface plasmon resonance (SPR) analysis, Fluorescence Resonance Energy Transfer (FRET) measurements, circular dichroism, and flow cytometry, that are used to evaluate the efficacy of bioaffinity nanoprobes. Additionally, it discusses advances in the development and application of biosensors in monitoring foodborne pathogens.

## 1. Introduction

Food safety has become a global concern, given the frequency and severity of foodborne disease outbreaks recently, and the grave effects associated with them. Most of these outbreaks are caused by foodborne pathogens, which are bacteria, viruses, and parasites that contaminate food, causing diseases and in some cases deaths. Some of the predominant foodborne pathogens include *Listeria monocytogenes*, *Campylobacter*, *Staphylococcus aureus*, *Salmonella typhimurium,* and *Escherichia coli* [1]. The advancement of food supply chains, which has become a giant network connected to all parts of the world, makes the spread of foodborne illnesses more rapid and the strain on socio-economic development disastrous. Although there have been great advancements in the methods used to detect pathogens, there are still increasing outbreaks of foodborne diseases showing that the methods of analysis are no longer enough.

The conventional culture-based methods are very cheap and easy to use but take up to several days to produce results and require extra biochemical or molecular tests to confirm that the species match the pathogen of interest, making them highly unsuitable for on-site detection [2]. Other methods have also been developed, such as the polymerase chain reaction (PCR), nucleic acid, and immunoassay-based methods. These methods are able to fix the time constraint of the culture-based methods but are usually expensive and require specific reagents, complicated sample pretreatment, and experienced personnel for the analysis, making the possibility of commercialization nearly impossible [3,4]. Hence, the urgent need for inexpensive, easy-to-use, but accurate, and rapid detection methods that do not require specialized expertise or equipment to run. Nanosensors, which are a product of biosensor technology incorporated into nanotechnology, are the newest and most advanced detection technology being developed by scientists [4,5].

A nanosensor is basically a compact analytical device with dimensions below 100 nanometers (nm) that detects the presence of biomolecules and nanoparticles or monitors physical and chemical parameters on a nanoscale. It can be a sensor with bioreceptor platforms or a biosensor with an added nanomaterial (NM) component to enhance sensitivity and efficiency [6,7,8]. Nanosensors incorporating additional nanomaterials offer advantages such as high absorption capacity, improved thermal stability, presence of surface-activated functional groups, large surface area, and excellent surface reactivity [9]. There are three main stages in all nanosensing methods: the recognition element, also known as the bioaffinity probe, detects the target by forming a complex with it. The transducer converts the target recognition event into a measurable signal, such as an electrochemical, colorimetric, impedimetric, or voltammetric signal. A third element manipulates the converted signal into a format that can be easily displayed or interpreted by the analyst [10]. The success of a nanosensor largely depends on the selectivity combined with the sensitivity and specificity of the recognition element, overall stability, limit of detection, cost, response time, shelf life, linearity, recyclability, and hysteresis [11].

Since foodborne pathogens are usually present in trace quantities, the bioreceptor of choice for the nanosensor must possess a high affinity towards the target to be able to detect it even in minute quantities. For this reason, biomolecules such as antibodies, aptamers, enzymes, cells, and proteins are preferred as bioaffinity probes in the nanosensors or nano-biosensors [12].

In order to analyze the real-time performance of these bioaffinity probes, certain techniques have been designed. These analytical approaches consist of Surface Plasmon Resonance (SPR), Circular Dichroism (CD), Flow Cytometry (FC), and Fluorescence Resonance Energy Transfer (FRET) measurements, and they are used to study the interactions between these bioaffinity probes and their targets.

## 2. Overview of Bioaffinity Nanoprobes

Bioaffinity nanoprobes are biomolecules that serve as bioreceptors in nanosensors. These molecules bind to the target to create a reaction that can be converted into an understandable signal to the analyst. These molecules include antibodies, aptamers, enzymes, and non-antibody binding proteins.

### 2.1. Antibodies

Antibodies, which are also known as immunoglobulins (Ig), are defined as large proteins having ‘Y’ shapes and are naturally produced in the body of an animal in response to antigens. Their main purpose is to serve as a defense mechanism of the immune system through phagocytosis, complement-mediated lysis, neutralization of infectivity, and antibody-dependent cellular cytotoxicity (ADCC) [13,14,15]. The structure of antibodies can be split into two; the first is the antigen-binding fragment (Fab), and the second is the constant or crystallizable fragment (Fc). The Fab region establishes the idiotype of the antibody and possesses affinity towards the target, transmitting a neutralizing effect to it once it binds to it. The Fc region controls other immune-associated activities including macrophage and complement binding, as well as defining the isotopes of the antibody [14,16]. All antibodies comprise four polypeptide chains held together by disulfide bonds. Two of these chains are heavy chains and the other two are light chains, which come together to create the Y-shaped structure [17,18]. Typically, antibodies are classified into five classes (IgA, IgD, IgE, IgG, and IgM) depending on the composition of their heavy chain constant region. These antibodies occur in the form of monomers (IgD, IgE, and IgG), dimers (IgA), or pentamers (IgM) [14,16]. There are two general types of antibodies, which are monoclonal antibodies and polyclonal antibodies. They have the same basic structure and function, but their differences have to do with their production and specificity. Table 1 provides a summary of the differences between monoclonal and polyclonal antibodies.

Scientists, for the past decades, have used antibodies as the predominant receptor for biosensors due to the natural antigen–antibody interaction. In the development of antibodies to be used as bioreceptors, there are three factors that are sought. The first is sensitivity, which is the ability of the antibody to be able to recognize and quantify the target molecule even when the concentrations are low. This is mostly a problem for foodborne pathogens since they usually occur in trace amounts and yet are very potent. The second is specificity, which is the ability to differentiate the pathogen strain even in the presence of other strains or pathogens. The last thing is high affinity, which is the ability to form a complex with its target that is strong enough to allow further analysis [20,21]. Selecting antibodies with all three characteristics is quite a task, but because monoclonal antibodies have very impressive specificities [22], scientists have been able to develop them for a myriad of foodborne pathogens over the years.

Recombinant technology has been infused into the development of antibodies to make the process more efficient. This technology allows the production of antibodies from synthetic antibody repertoires without the immunization of animals [23]. Plückthun and Skerra developed a method that uses vectors present in bacterial systems to create fully functional, completely folded antibody fragments [24], as opposed to the traditional method of generating the fragments by proteolytic cleavage alone [25].

Antibodies as bioaffinity nanoprobes have a wide variety of applications grouped into diagnostic and therapeutic medicine, agrobiotechnology, food safety, environmental protection, and many others. These antibodies cannot work independently, but they have to be incorporated into technologies such as colorimetric, electrochemical, voltammetric, and optical biosensors, or even conventional assays including the enzyme-linked immunosorbent assay (ELISA) to be able to function completely as a nanoprobe.

In the case of biosensors, their success depends partly on the ability to immobilize the antibody while maintaining their original activity. This immobilization step is so crucial because it affects the sensitivity and overall performance of the sensor, as well as the detection limit [17]. The antibodies can be immobilized on the solid sensor surface through methods such as:Adsorption using electrostatic or hydrophobic interactions;Entrapment;Covalent coupling using amine coupling, thiol coupling, or coupling through glycan moiety;Affinity: immobilization is performed through intermediate proteins such as in the case of avidin–biotin [13].

### 2.2. Enzymes

Enzymes are macromolecules that act as catalytic agents, meaning they accelerate the rate of biological or chemical reactions without being consumed or taking part in the reaction. They are usually proteins, but some RNA molecules called ribozymes have been found to possess catalytic abilities. They function by lowering the activation energy of the reaction as they stabilize the transition state. Some enzymes rely on other small non-protein molecules called coenzymes to function fully. Enzymes are far more efficient than most of the inorganic catalysts available. This is seen in their impressive specificity. Inorganic catalysts increase the rate of a bunch of chemical reactions in the system while enzymes target specific reactions. Even when they target more than one reaction, the reactions are usually of the same type and their reactants have similar structural traits. Because of the specificity of enzymes, they are able to selectively differentiate between substances (substrates or analytes), even those that are optical isomers.

As per the International Union of Biochemistry and Molecular Biology’s proposal, enzymes are classified into six primary groups, which include oxidoreductases that catalyze redox reactions, transferases that facilitate the transfer of atoms from a donor substrate to an acceptor, hydrolases that catalyze the breaking of bonds through the addition of water, lyases that cause the cleavage of bonds through methods other than hydrolysis, isomerases that catalyze the conversion of isomers, and ligases that catalyze the binding of molecules [26,27,28]. During a reaction, the enzyme binds to the substrate at a specific location to form a complex. These locations in the enzyme’s structure, called the active binding sites, take up only a small portion of the enzyme’s total size and are mostly filled with water in the absence of binding. They are often grooves and crevices that the substrate binds to in order for the reaction to be accelerated. Since enzymes are mostly proteins, they are made up of amino acids which form the primary, secondary, tertiary, and quaternary structures. The conformation of amino acids within the active sites plays a crucial role in stabilizing the specific binding of substrates and thereby determining the enzyme’s specificity [27,29,30]. Enzymes are isolated traditionally from natural sources; that is, from the organisms that provide an abundant or easily isolated source [31].

The unique characteristics of enzymes, that is, their ability to specifically recognize substrates and catalyze their transformation, giving rise to a signal, make enzymes a perfect bioaffinity probe, fit for use in a biosensor or nanosensor. Biosensors using enzymes as bioaffinity probes were the earliest biosensors to be developed. The amperometric enzyme electrode for glucose sensing utilizing a soluble enzyme electrode was first designed by Clark and Lyons in 1962 [32]. Since then, scientists have grown keen on the use of enzymes as bioaffinity probes in sensing even for the detection of foodborne pathogens. In the use of enzymes as bioaffinity probes, the analyte, specifically the foodborne pathogen, can be recognized by three means.

The first option is that the concentration of the enzyme can be estimated by measuring the catalytic transformation of the analyte, which is metabolized by the enzyme.Secondly, the enzyme is inhibited or activated by the analyte, hence the concentration of the analyte is proportional to the decrease in enzymatic product generation.The last option is by tracking the alteration in the characteristics of the enzyme.

The catalytic impact of the enzyme, upon which the theory of analyte detection is based, is also dependent on multiple characteristics, inclusive of the concentration of the analyte, pH, temperature, and the presence of either a competitive or non-competitive inhibitor [31]. The success of an enzyme in a nanosensor depends on its ability to be held on tightly or carried by a solid surface. The process of attaching the enzyme to the solid surface is referred to as immobilization. Just like antibodies, enzymes can be immobilized through entrapment, affinity attachment, and nonspecific covalent attachment [31]. Aside from using enzymes as the main bioaffinity probe in a sensor, they can also serve as labels in immunoassays (antibody-based biosensing), as in the case of alkaline phosphatase and horseradish peroxidase (HRP) [31,33]. Enzymes as bioaffinity nanoprobes have applications in several fields, including food safety, environmental monitoring, heavy metal detection, and health, and can be used for so long as they are not consumed.

Even though enzymes have excellent specificity and are perfect as nanoprobes, they have their own limitations which include it being expensive and difficult to find new active and efficient enzymes and difficulty in improving the sensitivity and adaptation to other functions [34].

### 2.3. Aptamers

The origin of the term “aptamer” can be traced back to the Latin word “aptus”, meaning “to fit”, and the Greek word “meros”, meaning “part” [35]. Aptamer is used to describe DNA or RNA oligonucleotides that are short and single-stranded, as well as peptides that can recognize their targets with exceptional affinity, selectivity, and specificity [36]. DNA and RNA aptamers were first unearthed in the year 1990 by two independent teams: Ellington and Szostak in the preparation of RNA molecules that targeted organic dyes [37] and Tuerk and Gold in T4 DNA polymerase [38]. In 1996, Colas et al. also introduced peptide aptamers, as they reported short structures of peptides with the ability to detect cyclin-dependent kinase 2 [39]. Aptamers are selected through a meticulous, repetitive procedure, consisting of a series of selection and amplification, popularly known as “Systematic Evolution of Ligands by Exponential Enrichment (SELEX).” This process involves three main stages, and they are shown in Figure 1.

The first step is the incubation of the target molecule with the library. In this step, the target is incubated into a large, random pool of about 10^15^ single-stranded nucleic acid sequences where there is an interaction between the target and nucleic acids [41].

The next step is the segregation of the nucleic acid–target complexes from the unbound sequences and the discarding of the unbound sequences [42].

The final step is the amplification of the sequences that formed the complexes by polymerase chain reaction (PCR) in the case of DNA or reverse transcription PCR (RT-PCR) for RNA. This amplified group of sequences becomes the new initial library for the next cycle [35].

The selection method is repeated until some oligonucleotide sequence(s) with exceptional specificity and selectivity are obtained. These become the selected aptamers, and it usually takes at least 8–15 rounds of SELEX to achieve, yet the whole process takes a few weeks [42]. Aptamers can be selected to detect a wide variety of targets including proteins, bacteria, viruses, protozoa, small chemicals, metal ions, antibiotics, parts of cells, and even whole cells [43]. They are also applied in so many fields such as the medical sector for diagnosis and therapy, food quality, environmental safety, research, and bioanalysis. Aptamers have become a highly sought-after choice in the development of biosensors due to the mouthwatering advantages they possess. Some of these advantages are excellent affinity, sensitivity, and selectivity towards targets, low cost of production, shorter production time, low toxicity, easy modification, ability to easily permeate tissues due to small size, stability in extreme conditions, and the ability to retain their original conformation when favorable conditions are restored [44,45,46]. Aptamers can be described as a prominent successor of antibodies in bioanalytics and nanosensor development as they provide solutions for most of their limitations and a competitive affinity and limit of detection of targets.

### 2.4. Other Bioaffinity Nanoprobes

#### 2.4.1. Non-Antibody Binding Proteins

Non-antibody binding proteins, also known as synthetic binding proteins, are proteins with a non-immunoglobulin fold generated by non-antibody scaffolds. These scaffold domains are obtained by creating a random library through targeted mutagenesis in a loop region or another acceptable surface area. Variants are then selected against a specific target using phage display or other molecular selection methods [47]. While several protein scaffold options have been proposed, only a few have been proven to provide specificities for various target types and offer practical advantages. The scaffold domains that have been found to produce these proteins include Anticalins, Lipocalin, Sso7d protein, Darpins, Fibronectin type 3, Affibodies, and ankyrin repeat protein [47,48]. Non-antibody binding proteins offer advantages such as low molecular weight, which facilitates tissue penetration; high thermal stability, with approximately 70% of the available scaffolds having denaturation temperatures between 37 and 120 °C; ease and cost-effectiveness of production compared to antibodies; longer shelf life; and robustness [49]. These scaffold proteins enable the generation of chemically consistent proteins that can be tailored to detect various analytes without significantly affecting the biosensor configuration, while also enhancing the packing density of the recognition element [50]. These benefits, combined with their ease of expression, justify their use as a viable alternative to traditional antibodies or their recombinant fragments [47].

#### 2.4.2. Molecularly Imprinted Polymers

Molecular imprinting is a template-guided process that creates selective pockets within a three-dimensional polymeric matrix. By removing the template from the polymer, functional porous materials with high-affinity binding sites, known as molecularly imprinted polymers (MIPs), are obtained. These binding pockets have configurations and functionalities that match those of their target molecules [51,52]. The synthesis of MIPs involves a three-step process:Incubation: Monomers are incubated with a dummy, epitope, or template molecule, which facilitates the formation and stabilization of non-covalent interactions between the functional monomers and the template.Polymer Formation: The polymer is formed around the template with the help of cross-linkers, resulting in the creation of a network structure.Template Removal: Suitable solvents are used to remove the templates, leaving behind specific binding sites that are complementary to the template molecule [53,54].

MIPs offer several advantages over antibodies, making them a promising alternative. These advantages include structure predictability, chemical and thermal stability, longer shelf life, cost-effectiveness and ease of production, minimal batch-to-batch variation during mass production, and high sensitivity. Due to these properties, MIPs find applications in various fields such as food safety, environmental science, therapeutics, and more [51,53,55]. The ongoing research and development in molecular imprinting techniques continue to enhance the selectivity, stability, and sensitivity of MIPs, further expanding their potential applications in various scientific and technological fields.

## 3. Analytical Approaches for the Assessment of Bioaffinity Nanoprobes

### 3.1. SPR

Surface plasmon resonance (SPR) is one of the most prominent sensitive and qualitative, label-free techniques used to monitor binding events and to measure the relations between biomolecules such as protein and protein, protein and DNA/RNA, enzyme–substrate/inhibitor, and receptor–drug [56,57]. It is a spectroscopic method that measures the refractive index changes very at the surface of thin metals such as gold, silver, and aluminum films as a result of biomolecular interactions. Generally, when incident light strikes the metal surface at a given angle (incidence angle), the photons induce an excitation of the free electrons in the surface coating of the metal, causing them to oscillate. The movement of the electrons is called plasmon and it is always parallel to the surface of the metal [56]. A typical SPR equipment comprises a source of monochromatic polarized light and a thin film of metal (most often gold) supported by a glass prism in combination with a photodetector, which is represented in Figure 2.

When the polarized light goes through the glass prism, an evanescent wave that passes through the thin film of metal is generated from the internally reflected light. If the intensity of the reflected light is monitored with time in relation to the angle of incidence, a minimum reflected light will be achieved at an incident angle referred to as an SPR angle. This SPR angle depends on certain optical characteristics of the system such as the refractive index within close proximity of the metal film surface [59]. The sample solution, containing the target molecule most of the time, flows across the SPR surface after the bioaffinity probe (antibodies, aptamers, enzymes) has been immobilized unto the solid surface [60]. When there is any form of interaction at the surface of the metal, an alteration in the refractive index will be triggered, resulting in a change in the SPR angle and producing a signal that can be detected [61,62]. The amount of analyte that is bound to the bioaffinity probe is measured by observing the intensity of the reflected light or the shifts in the resonance angle, making it a real-time analysis method [63]. The gold metal and its glass support make up the SPR sensor chip and it is upon this chip that ligands, which in this sense are the bioaffinity probes, are immobilized, sometimes with the help of a polymer matrix.

There are so many different chemical mechanisms that are used for immobilization. Some of them are aldehyde, amino, carboxyl, hydroxyl, and thiol group coupling. Sometimes immobilization cannot be performed directly; hence, certain molecules can be used as capture surfaces to enhance the immobilization process. These are biotin, histidine-tagged, and glutathione-S-transferase fusion proteins [57,61]. Immobilization of the bio affinity probe on the sensor surface is by far the most important step of the SPR analysis since the success of the binding analysis somewhat depends on the response generated by the immobilized ligand [64]. The SPR equipment generates output data in the form of a sensorgram, which is a plot of response units (RU) with respect to time, and Figure 3 gives a pictural view of what a typical sensorgram looks like.

The sensorgram starts with a baseline, indicative of the response before the start of any form of interaction. When the analyte solution flows close to the surface of the sensor, the analysis enters an association phase which shows the binding of the analyte to the immobilized ligand. This is seen in the steady rise in response on the sensorgram to a point where the complex attains equilibrium and the curve flattens out. Right after the equilibrium phase, a drop in response is observed, which indicates the dissociation phase. This is the stage where the ligand–analyte complex separates, as the SPR system stops the flow of the analyte solution and switches to the flow of a running buffer. More often than not, the complex does not dissociate completely and a regeneration solution, which is usually a mild alkaline or acidic solution, is used to regenerate the sensor surface for subsequent analysis [57,66,67]. Most of the SPR equipment available, such as the Biacore equipment, is able to generate a table of the raw data, the association constant (ka), the dissociation constant (kd), and, most importantly, the equilibrium dissociation constant (Kd), together with some statistical models to fit the data. The equilibrium dissociation constant is, however, the star of the show, because it gives a clear picture of the kinetics of the interaction and binding affinity of the bioaffinity nanoprobe used as the ligand [68]. A very small value of Kd, recorded in the nanomolar to picomolar range, shows that the ligand has a high affinity towards its target and hence will be an excellent bioaffinity probe when used as a biosensor for the detection of pathogens. It shows how easily the ligand detects the analyte in low concentrations, how tightly the ligand binds to the analyte, and how difficult it is to separate the complex [69]. These are the qualities required in a great bioaffinity nanoprobe. The SPR has the advantages of providing an automated and rapid alternative to cell-based assays, the lack of a need for reporter molecules such as fluorochromes or radioisotopes for a binding signal to be recorded, hence the biomolecular interaction is evaluated in real-time, and the ligand and analyte involved in the interaction do not lose their conformational integrity [70]. There are also a few limitations which include degrading the sensor surface due to harsh conditions of regeneration, and the immobilization of a sufficient amount of ligand on the sensor surface must also be successful [71].

### 3.2. FRET

Förster resonance energy transfer (FRET), which is also widely known as fluorescence resonance energy transfer, is a technique whereby excited state fluorophores non-radiatively transfer electromagnetic energy to other fluorophores which are about 1–10 nm away. The fluorophore involved in the transfer is termed the donor, and the one receiving the energy (often ground state level), is termed the acceptor [72,73,74,75]. Energy transfer is facilitated through the energetic coupling of transition dipoles between the two fluorophores and can only occur when there is a spectral overlap between the emission spectrum of the donor and the excitation spectrum of the acceptor [74,76,77]. The transfer eventually leads to the donating fluorophore entering the ground state level and the fluorophore accepting becomes excited [77]. During FRET, the likelihood of an excited donor fluorophore returning to the ground state is commonly known as the transfer efficiency (*E*). This efficiency is dependent on the physical distance from the center of the donor to the center of the acceptor of the FRET pair, “*r*”, as well as the characteristic Förster distance, also known as the quenching radius, “*R_o_*”. *R_o_* is typically in the range of 2–8 nm and it is bound by several factors shown by Equation (2). The connection between the transfer efficiency and the two important distances is shown in Equation (1) [75,78,79].
(1)$$E=\frac{R_{o}^{6}}{R_{o}^{6}+r^{6}},$$
(2)$$R_{o}^{6}=8.79\times 10^{-25}k^{2}\eta^{-4}J\left(\lambda \right)\phi ,$$

Looking at the first equation, it can be inferred that at *r* = *R_o_*, *E* = ½ and *R_o_* defines the tiny distance (nm) that exists between the two fluorophores at the point where half of the entire donor relaxation processes occur by transferring energy to the acceptor [77,80]. The magnitude of *R_o_* depends on the orientation (*k*^2^), the medium’s refractive index (*η*), and the donor’s quantum yield (*ϕ*), in addition to the degree at which the spectra of the donor and acceptor overlap (*J*(*λ*)). The number of unquenched donor fluorophores represents the rate at which energy is transferred between the donor and acceptor [75,77]. Figure 4 shows a schematic diagram of the fundamentals of FRET.

FRET is preferred over other options of interaction analysis because it has very minimal restrictions and can even be used within a living cell. The major requirement is the ability of light to be delivered to and collected from the sample, but generally, only simple benchtop equipment is needed [74]. It is perfect for biomolecular interaction analysis because the majority of the biomolecules are in the nanoscale range and FRET is also viable in that same range. The nanometric distance measurements can also serve as a ‘molecular gauge’ for biomolecular structure analysis [82]. FRET is versatile enough to be applied in diverse areas of biomolecular research, but it is also sensitive to environmental conditions such as solvent pH, viscosity, polarity, and many others [83]. FRET analysis can be carried out either with a single FRET or the multiplexed FRET methods. Multiplexed FRET methods offer much more advantages including simultaneous analysis of multiple analytes, analysis of intermolecular and intramolecular interactions, and monitoring of coinciding biomolecular events [84].

The major disadvantage of FRET is that it does not report directly and specifically on the interactions between biomolecules, it only measures the donor–acceptor proximity and stoichiometry, hence the conclusions drawn are not strong enough without additional data or information [74].

### 3.3. CD

Circular dichroism (CD) is an absorption spectroscopy used to investigate optical isomerism and secondary structures of molecules by taking the difference in absorptions of the left and right lights that are circularly polarized by chiral molecules [85,86]. Chiral molecules are those that are not superimposable on their mirror images and hence exhibit optical activity as an effect [87]. Upon the passage of light through a chromophore solution, the light may either be absorbed or refracted. Absorption is quantified by the molar extinction coefficient, ‘epsilon’. Molecules that are active optically have unique molar extinction coefficients for the two different circularly polarized lights. The deviation between the absorbance of the two different circularly polarized lights can be represented by a constant described by the Lambert–Beer law as delta A. The difference between the delta A of the two components or molecules is the measure of the Circular Dichroism. Numerous articles have extensively explained the calculations involved [88]. Electronic CD is produced mainly by molecules whose chromophores can absorb light in the ultraviolet (UV) and also the visible spectral territories and is used to study charge transfer transitions in metal–protein complexes. Vibrational CD, on the other hand, is generated in the infrared (IR) spectral region and it is useful in the analysis of the structure of organic molecules of relatively small size, such as proteins and DNA [86,87]. Figure 5 is a schematic representation of the arrangement in a CD equipment.

Applications of CD in Biomolecular studies are vast, but it is mostly used for the comparison and characterization of protein secondary structures. It provides an efficient method to study the effect a mutation or change in environmental conditions of the protein may have on the overall structure [90]. CD can also be applied in the analysis of the interaction between molecules such as DNA and DNA-binding ligands. Many ligands that can bind to DNA are not chiral, and hence, are not active optically. However, when they interact with DNA, an induced CD (ICD) signal can be achieved by the ligand by virtue of the joining of the moments of electric transition of the ligand to that of the bases of the DNA. When ICD signals are observed within the absorption bands of the non-chiral ligand, it is a clear indication of binding between the ligand and the DNA [91]. Even though CD provides much lower resolution than other analysis methods such as X-ray crystallography and nuclear magnetic resonance, it has certain advantages that cannot be overlooked. Analysis can be very rapid and inexpensive, only small amounts of sample are required; CD is not affected by the molecule’s molecular weight [91,92].

Beyond the advantages of CD, it has a few limitations. For example, specifying the ideal parameters necessary for great CD results in the instrument or experimental procedure is quite challenging and the data obtained are difficult to interpret or make sense of [87].

### 3.4. FC

Flow cytometry (FC) is a rapid detection and characterization technique used for biomolecules in a salt-dominated solution as they flow through either single or multiple lasers [93]. The word cytometry in the name literally means “cell measurement”, as it was originally designed to measure mammalian cells suspended in a flowing stream [94]. FC is able to provide information on the cell number, type, cell physiology, cell viability, susceptibility, genetic identity, and important metabolic parameters on the level of a single cell, and even whole eukaryotic cells across large populations [95]. A flow cytometer basically comprises a source of light, an optical bench, a fluidic system, electronics, and a computer to control the equipment [96]. The sample flows in single file by the action of isotonic sheath fluid in the fluid system and is exposed to a light source or sources. Light signals generated by the light sources are guided by the optical system towards photodetectors, which then transform the light into electronic signals that are stored for later analysis. Because the fluidic system is in the middle of the cytometer, the cell streams are centrally placed, ensuring that the brightness of all the cells is similar. This way, any variation in the value of signals emitted from the cells will reflect actual biological differences [97]. The illumination process produces both fluorescent and non-fluorescent signals. These signals are analyzed by optically joining the signal to a system of detection, which is made up of filters that are linked to a photodetector. The photodetectors’ number and configuration permit the concurrent evaluation of many different parameters for a given cell. The electronics part of the cytometer provides a system that converts the analog light signals coming through the photodetectors to digital signals that can be read and stored in the computer [97]. Most flow cytometers available for commercial use have a principal laser, which is an argon-ion laser set at 488 nm. Modern lasers at different wavelengths comprising ultraviolet (350 nm), red (635 nm), violet (405 nm), blue (488 nm), yellow (560 nm), and green (532 nm) allowing the instantaneous use of several fluorophores, with varying excitation needs, are becoming common as well [93,98,99].

Flow cytometers could be either imaging flow cytometers (IFC), which combine the conventional FC and fluorescence microscopy for sample morphology analysis, along with multi-parameter fluorescence [100], or mass cytometers, which integrate time-of-flight mass spectroscopy with FC [101,102]. The advantages of FC that make them so attractive to biosensing are the facts that they are rapid, they can probe a huge number of cells (up to 10^6^–10^8^ cells per sample), they can measure fluorescence intensity quantitatively, they can identify pathogens in complex matrices such as food without target enrichment or isolation [103,104]. There are certain limitations of FC that impede the full-scale use of the technique in biomolecular assays; the samples need to be in a single-cell suspension, it is difficult to find the right combinations of antibodies and fluorophores with minimal spectral overlapping, and extra care is needed in the interpretation of FC data. Figure 6 shows a graphical representation of how flow cytometers function.

## 4. Application of Bioaffinity Nanoprobes in Food Biosensing

### 4.1. Electrochemical Sensors

Electrochemical biosensors are a product of biological and electronic technology, whereby a biological recognition element is coupled with conducting and/or semi-conducting materials known as electrodes. Some of the biological recognition elements have been discussed extensively in the bioaffinity nanoprobes section, including antibodies, aptamers, enzymes, and other peptides. Figure 7 is a schematic representation of an electrochemical biosensor.

In an electrochemical biosensor, an electrochemical method, usually involving an electrode and an electrolyte solution containing the analyte, transforms the chemical energy corresponding to the binding activity between the target and bioaffinity nanoprobe into electrical energy [107,108]. Electrochemical biosensors utilize different transduction methods which include electrochemical impedance spectroscopy (EIS), amperometry (I-t), and voltammetry (cyclic, differential pulse, linear sweep, square wave) [109,110]. Voltametric electrochemical biosensors have become one of the most versatile detection methods due to their lower noise tendency. These biosensors measure current in a steady potential, controlled by the working electrode, and the target concentration is obtained by observing the highest current intensity.

EIS is a frequency domain system that can measure a wide range of frequencies, providing more kinetic and structural information about the electrode interface than traditional electrochemical biosensors. In this type of biosensor, the interaction between the target and bioaffinity nanoprobe causes changes in the electric field, affecting the impedance values [110]. Electrochemical biosensors are such an attractive choice of pathogen detection technique, especially in food, because they offer a rapid, accurate, sensitive, inexpensive detection mechanism, requiring very small sample quantities. Nanomaterials and nanocomposites are commonly used to enhance the sensitivity of electrochemical biosensors. Additionally, these biosensors can be integrated with microfluidic systems to create compact and efficient devices with multiple functionalities in a single platform. Electrochemical biosensors have proven to be successful in detecting a wide range of pathogens and disease biomarkers. Their applications span research, diagnostics, therapeutics, food safety, and environmental monitoring [111,112,113].

Bekir et al. introduced a highly sensitive electrochemical immunosensor for detecting stressed and resuscitated pathogenic *Staphylococcus aureus*. The interaction was described by voltammetry, along with impedance spectroscopy. In the dynamic concentration span of 10^1^ to 10^7^ CFU/mL, an incredible linear response in addition to a low detection limit was recorded. The results were reproducible, indicating the viability of the system [114].

A label-free EIS was designed by Dong et al. based on gold nanoparticles and a poly(amidoamine)-multiwalled carbon nanotube-chitosan (AuNPs/PAMAM-MWCNT-Chi) nanocomposite film-altered glass carbon electrode for detecting *Salmonella typhimurium*. Bacteria in the linear range of 10^3^ to 10^7^ CFU/mL were recognized by the sensor, recording a limit of detection (LOD) of 5.0 × 10^2^ CFU/mL [115].

In 2018, a technique was presented by Helali et al. for detecting *Escherichia coli* in chicken by EIS and SPR imaging techniques. The detection limit obtained was 10^3^ CFU/mL [116].

Shimaa et al. also documented a new electrochemical biosensor for the concurrent detection of *Listeria monocytogenes*, as well as *Staphylococcus aureus*. They recorded outstanding sensitivities with LODs of 9 CFU/mL in the case of *Listeria monocytogenes* and 3 CFU/mL for *Staphylococcus aureus* [117].

### 4.2. Colorimetric Sensors

Colorimetry is the quantification of ultraviolet-visible (UV-vis) light that is being absorbed or reflected by a medium [118]. Colorimetric sensors are described as a class of optical sensors (sensors that use light in the infrared, visible, or ultraviolet region to analyze chemical or biological interactions), that show a single, double, or multiple change of color when a target molecule is recognized. They are easy to use, portable, cheap, and offer sensitive and selective on-site or in situ applications [119]. Colorimetric biosensors can be used for the detection of a specific analyte in a liquid sample through color changes that occur as a result of the interactions between the target and the bioaffinity nanoprobe, usually with the assistance of a color reagent, and this change in color is observable with the human eye or with very simple, compact optical detectors for quantitative analysis [120].

A colorimetric sensor consists of a source of light, a device for the selection of wavelengths such as filters or monochromators, a cell in which variations in the light absorbed or emitted in the presence of the target molecule can happen, and a sensitive detector [121]. Different types of colorimetric assays have been developed over the years for the application of pathogen detection and these include loop-mediated isothermal amplification (LAMP), polymerized polydiacetylene, gene expression reaction, and so on [120]. Several colorimetric sensors rely on the traditional three-channel visible range, which corresponds to the wavelength ranges partially overlapping with red, green, and blue. The use of many different channels with a smaller spectral range for every one of them is referred to as hyperspectral imaging. This approach can also be employed in colorimetric sensors. Colorimetric sensors can incorporate a broad range of wavelengths, including non-visible wavelengths, starting from near-infrared to ultraviolet, by using hundreds of color channels. This is known as full spectrophotometry.

In order to make the data analysis and instrumentation easier, the analysis of spectra is performed mostly at only a few discrete wavelengths or just by choosing the maximum points in the UV-vis spectra [121]. Figure 8 is a graphical representation of a colorimetric assay for the detection of staphylococcus aureus.

The major limitation of simple colorimetric sensors is low sensitivity, as it is difficult to transform detectable signals into specific color readouts. To overcome this limitation, a couple of nanomaterials such as graphene oxide (GO), gold nanoparticles (AuNPs), magnetic NPs, carbon nanotubes (CNTs), conjugated polymers, and cerium oxide NPs, have been developed and incorporated into the colorimetric assays [123].

Zhang et al. created a rapid, specific colorimetric biosensor for detecting *Listeria monocytogenes*, by using a vancomycin-conjugated, Fe_3_O_4_ NP cluster-improved aptamer as the bioaffinity nanoprobe. The system was a success, with a wide linear range given as 5.4 × 10^3^–10^8^ CFU/mL and a 5.4 × 10^3^ CFU/mL visible detection limit [124].

A specific, rapid, colorimetric aptasensing method for detecting *Salmonella* (S.) *typhimurium* was designed by Yuan et al. The sensitivity reached 7 CFU/mL using the human eye. The system could be adjusted for the concurrent recognition of *S. Typhimurium* and other foodborne pathogens [125].

Ren et al. described the development of a lateral flow sensor that utilizes plasmonic enhancement to significantly increase the colorimetric signal. The sensor relies on liposome-encapsulated reagents that induce the aggregation of gold nanoparticles (AuNPs). The procedure optimized the performance of the system for detecting *Escherichia coli* O157:H7 and made it better by 1000-fold. This led to a sensitivity of 600 CFU/mL with the naked eye in apple juice [126].

### 4.3. Optical Sensors

Optical sensors quantify the interaction between a receptor and an analyte by assessing a specific aspect of the reaction as an observable optical signal [127]. The majority of optical sensors measure changes in the sensor’s surface properties when the analyte binds to the sensing layer through adsorption or complex formation [127]. Optical biosensors combine biological selectivity with modern micro- and optoelectronics, finding applications in areas such as food safety, therapeutics, and environmental monitoring [128]. There are various types of optical sensors, including colorimetric, chemiluminescence, Fourier-transform infrared (FTIR) spectroscopy, matrix-assisted laser desorption ionization time-of-flight mass spectroscopy (MALDI-TOF-MS), fluorescence, surface plasmon resonance (SPR), Raman spectroscopy, and evanescent field optical fiber [127,129]. Optical sensors are preferred for foodborne pathogen detection due to their ability to detect targets in complex food matrices with minimal sample treatment. They offer high sensitivity and specificity, ease of use, cost-effectiveness, label-free detection, compactness, and minimal invasiveness [128].

Masdor et al. developed three distinct immunoassays, that is, direct, sandwich with gold nanoparticles (AuNPs) and sandwich for the detection of *Campylobacter* (C.) *jejuni* on the SPR equipment. In the direct analysis, the polyclonal antibody against *C. jejuni* was initially attached to the surface to serve as the capturing antibody. *C. jejuni* cells in different concentrations were subsequently introduced to the ligand, and the resulting response from the interaction was documented in response units (RU). The maximum response was observed at a concentration of 1 × 10^9^ CFU/mL and a response of 144.34 RU. The determined limit of detection (LOD) value was 8 × 10^6^ CFU/mL. In the sandwich assay, a capture antibody and a mouse control antibody were used. The greatest response was achieved at a concentration of 1 × 10^9^ CFU/mL, with a binding response of 131.5 RU. The calculated limit of detection (LOD) value was 4 × 10^4^ CFU/mL, and a strong coefficient of correlation of 0.997 was observed. This represents a notable improvement compared to the previous direct format, which had an LOD of 8 × 10^6^ CFU/mL. In the case of the sandwich assay with amplification of signal using AuNPs, the antibody-linked AuNPs were introduced over the detected bacteria, which increased the refractive index and, in turn, enhanced the binding response. The maximum response was observed at a concentration of 1 × 10^9^ CFU/mL, with an interaction response of 96.6 RU. The calculated LOD was 8 × 10^5^ CFU/mL, and a satisfactory coefficient of correlation of 0.998 was noted. The sandwich assay outperformed the others, while the direct assay was the least effective. Comprehensive cross-reactivity studies against various foodborne pathogens revealed minimal non-specific binding, making this assay even more specific than the other available methods [130].

Sanati and colleagues used an asymmetric, Vernier-type double-stage ring resonator (DSRR) integrated with a plasmonic slot waveguide for the identification of Escherichia (E.) coli K12 bacteria in potable water. The efficiency of the sensor was evaluated across a range of liquid environments, and the capacity of the DSRR sensor for the label-free identification of *E. coli* K12 at visible wavelengths was established. The suggested sensor delivers a high sensitivity value of 480 nm/RIU and an impressively low detection limit reaching down to 3.33 × 10^−5^ RIU. This makes the sensor a strong contender for swift and high-definition identification of *E. coli* bacteria in food items [131].

Kim and colleagues developed a *Salmonella* sensing platform utilizing retroreflective Janus microparticles (RJP) along with a simple optical system. In contrast to traditional fluorescence-based *Salmonella* detection methods, the RJP-based platform does not necessitate intricate optical tools, as RJPs can be visualized using a CMOS camera and a standard white LED. The system allows for highly sensitive and quantifiable detection of *Salmonella*. Moreover, the system exhibited high selectivity for invA by employing oligonucleotides with mismatched sequences. The invA gene encodes a protein that facilitates *Salmonella* invasion via a type 3 secretion system. Utilizing this system, concentrations of *Salmonella* varying from 0 to 100 nM were scrutinized with exceptional selectivity and sensitivity, achieving a detection limit of 2.48 pM [132].

### 4.4. Piezoelectric Sensors

Piezoelectric sensors are mass-sensitive sensors that are able to detect targets or analytes using a transduction mechanism that depends on small changes in mass. The technique employed for pathogen detection in this approach is contingent on mass evaluation via piezoelectric crystals. These crystals have the capacity to vibrate at a specific frequency when subjected to an electrical signal of a corresponding frequency. As a result, the vibration frequency is determined by both the crystal’s mass and the frequency of the electrical signal applied [133,134]. In the case of foodborne pathogen detection, when the mass increases due to the interaction with target pathogens, the crystal’s oscillation frequency changes, and the resulting shift can be measured electrically. This measurement is then used to calculate the additional crystal mass [135]. The two primary categories of mass-sensitive biosensors include surface acoustic wave devices and quartz crystal microbalance devices, which are also known as bulk wave devices [133]. Piezoelectric biosensors offer advantages such as low cost, simplicity, user-friendliness, and direct label-free analysis while maintaining consistent reliability and improved sensitivity [134]. Piezoelectric biosensors utilizing quartz crystal microbalance being the most common type have been customized with various antibodies and other bioreceptors for the recognition of foodborne and waterborne pathogens. These include *Salmonella*, *Escherichia coli*, protozoa, *Shigella*, influenza A and B viruses, *Campylobacter*, *Yersinia*, and *Vibrio* [136].

In a study by Lian and colleagues, they engineered an innovative sensor that integrates graphene, an aptamer, and interdigitated gold electrode (IDE) for the rapid and targeted recognition of *Staphylococcus aureus* (*S. aureus*). The biological recognition element in this process is the *S. aureus* aptamer. A compound known as 4-Mercaptobenzene-diazonium tetrafluoroborate (MBDT) salt served as the molecular bridge, chemically binding graphene to the IDE. These electrodes were, in turn, linked to a series electrode piezoelectric quartz crystal (SPQC). The *S. aureus* aptamers were then affixed onto the graphene via π–π stacking of DNA bases. When *S. aureus* is present, it specifically binds to the aptamer, leading to an interaction of the DNA bases with the aptamer and its subsequent release from the graphene surface. This action modifies the electrical characteristics of the electrode surface, thereby leading to a shift in the SPQC’s oscillator frequency. The detection process takes only 60 min to complete. The sensor displayed a proportional correlation between shifts in resonance frequency and the range of bacterial concentrations from 4.1 × 10^1^ to 4.1 × 10^5^ cfu/mL. Moreover, it demonstrated a sensitivity with a lower detection limit at 41 cfu/mL [137].

Sharma et al. were able to detect *Listeria monocytogenes* (LM), an infectious bacterium, at the infection dose threshold of 10^3^/mL within an hour in both a buffer solution and milk. This was achieved using a unique asymmetrically anchored cantilever sensor and a commercially procured antibody. To validate the responses of the sensor, a secondary antibody-binding phase, akin to sandwich ELISA tests, was utilized for signal boost and the minimization of false negatives. Through the incorporation of a tertiary antibody-binding phase, the team was successful in detecting LM at concentrations as low as 10^2^/mL, a level significantly below the infection dose (<1000 cells) for LM [138].

Table 2 summarizes more of the applications of nanoprobes, taking into consideration the sensor types, targets, limits of detection, and samples tested.

### 4.5. Newer Technologies—Microfluidic Detection Methods

The cutting-edge approach to pathogen detection utilizes compact, integrated biosensing technologies, delivering dependable, sensitive, economical, and quick detection without the necessity for intricate equipment. Microfluidics is a versatile platform engineered for the streamlining, consolidating, and miniaturizing of devices, making it particularly well-suited for electrochemical, biomedical, and biochemical applications. It is the basis of point-of-care (POC) detection, of which paper-based and lab-on-chips (LOC) are the most outstanding technologies. Several applications of LOCs or microfluidics in foodborne pathogen detection have been covered extensively in the literature.

Sun et al. developed a micro-spot paper-based analytical device (μPADs) by the combination of a PVC pad and filter paper. The detection method relied on the observation of a color shift (from colorless to indigo) upon the interaction of a unique enzyme linked to the *Cronobacter* spp. under examination with a chromogenic substrate. By fine-tuning the enrichment steps, the technique permits an analysis duration of 10 h or fewer and is able to identify living bacteria on the injected sample surface in concentrations as low as 10^1^ CFU/cm^2^. This work showed that the production technique is innovative, straightforward, highly reproducible (having an RSD below 5%), and inexpensive (below $0.15 for every micro-spot) [149].

A colorimetric paper-based analytical device (PAD) combined with immunomagnetic separation (IMS) was developed for recognizing *Salmonella* (S.) *typhimurium* by Srisa-Art et al. IMS utilized coated anti-*Salmonella* magnetic beads to detect, separate, and preconcentrate bacteria from samples before testing on paper. A sandwich antibody-based assay was integrated into the process, employing β-galactosidase (β-gal) to be the enzyme of detection for direct *S. Typhimurium* detection after IMS. The antibody and enzyme complex enabled a colorimetric assay using chlorophenol red-β-d-galactopyranoside (CPRG) for bacteria detection. The procedure showed high specificity to *S. Typhimurium* with no interference from other pathogens such as *E. coli*. Without pre-enrichment, the detection limit of *S. Typhimurium* in culture solution was found to be 10^2^ CFU/mL. The developed system was put to use to identify *S. Typhimurium* in fecal samples from starlings that had been inoculated, as well as in whole milk. The system showed detection thresholds of 10^5^ CFU/g in the bird feces and 10^3^ CFU/mL in the milk. Notably, this represents the first documented use of a paper-based technique for detecting *S. Typhimurium* in such samples [150].

Smartphones have become valuable and readily available tools for diagnostics, removing the need for costly signal readers. Combining biosensing technology and digital communication systems, these devices offer immense potential for detecting pathogens in various areas, such as water, food, plant nurseries, medical, environmental, and wastewater. The data collected from the analysis can be easily stored, compared, and transferred between systems, making smartphones an efficient and affordable solution for diagnostics [151].

Cheng et al. reported a nanosensor that employed platinum–palladium (Pt-Pd) nanoparticles as signal boosters in a dual lateral flow immunoassay (LFIA) system, which was combined with a device based on smartphones, for the concurrent detection of *Salmonella Enteritidis* and *Escherichia coli* O157:H7. Following optimization, the detection limits were found to be around 20 CFU/mL for *Salmonella Enteritidis* and roughly 34 CFU/mL for *E. coli* O157:H7. The recovery rates for the dual LFIA method ranged from 91.44% to 117.00%, indicating its effectiveness in identifying live bacteria present in food samples [152].

Jung and colleagues also utilized the high-resolution camera, steady source of light, and computational aptitude of a smartphone to devise a method that objectively and accurately determines bacterial cell concentrations in food samples, using a regression model based on the intensity of the color of the test lines. They designed a 3D-printed sample container compatible with standard lateral flow assays and developed a custom Android app to extract cell concentration data from color intensity measurements. Tests using *Escherichia coli* O157:H7 as a representative organism showed that the smartphone-based procedure could detect concentrations between 10^4^ and 10^5^ CFU/mL in both spinach and ground beef samples [153]. Many examples of the applications of these newer detection techniques have been mentioned extensively in various articles [151,154,155,156].

## 5. Future Perspective

It is a fact that a broad range of bioaffinity nanoprobes have been selected or produced for foodborne pathogen detection and with that, many modes of analysis for the success or progress of these nanoprobes have been reported or enhanced. These bioaffinity nanoprobes, especially aptamers, have made great strides in research into biosensing, diagnostics, and therapeutics but have gained very little success in commercialization. The future for bioaffinity nanoprobes, in general, is focusing on modifying them to meet these criteria: cost-effectiveness, accuracy and precision, sensitivity and selectivity, as well as operation [157].

In the context of developing biosensors for the recognition of foodborne pathogens, it is essential to ensure that biosensors have the capability to specifically detect the target pathogen and provide explicit results that give the analyst or user certainty or confidence in the results, as pathogens in food are usually in trace amounts. This depends largely on the selectivity, sensitivity, and specificity of the bioaffinity nanoprobes. Many nanomaterials have been incorporated into biosensing to assist with this aspect, and many more will be developed in the future. Given the complexity of food structures, there is a need to look into the development of novel bioaffinity probes and the enhancement of the existing options to achieve the ultimate goal of high sensitivity and efficiency in the detection (LOD < 10^2^ CFU/mL) of pathogens even with the ever-changing trends in food processing, distribution, and consumption.

One of the key challenges of foodborne pathogen detection methods that currently exist is the need for specific sample preparation protocols which require sample purification and enrichment prior to the analysis. To be able to use biosensors effectively for rapid, point-of-care, or in situ applications, there is a need to develop analysis methods that can function with extremely small sample quantities and minimal sample preparation. The microfluidic chip technology has been a great innovation in miniaturized detection systems, offering the advantages of consumption of minimal samples and reagents, simultaneous analysis, controllable liquid flow, and an incredibly decreased analysis time. This technology, when improved and incorporated into detection techniques, will be helpful in the future [157]. Another reason why the commercialization of systems of analysis utilizing bioaffinity nanoprobes is lagging is the cost associated with the development of the sensors and the display platforms.

Paper-based biosensors have been introduced as the alternative to traditional biosensors because they are cheap, portable, and simple to use. This is very good for in situ pathogen detection even in developing countries that have limited resources. Smartphones, as display platforms for the analysis of detection results, have also been suggested by researchers for the sensing of foodborne pathogens. Given the portability, high camera quality, and availability of smartphones, they will be an effective tool if incorporated into biosensing on a larger scale together with cheaper sensing techniques such as paper-based biosensors.

Multimodal detection, offering a promising strategy for the comprehensive and reliable identification of foodborne pathogens, is a vital future research endeavor. By combining multiple sensing modalities, such as optical, electrochemical, and molecular techniques, a synergistic effect can be achieved, leading to enhanced sensitivity and specificity. For instance, a multimodal biosensor can integrate optical detection methods, such as surface plasmon resonance or fluorescence, with electrochemical transduction for simultaneous measurement of multiple target analytes. This multimodal approach enables the detection of pathogens through different recognition mechanisms, increasing the likelihood of accurate identification even in complex food matrices. Moreover, the combination of different techniques can provide complementary information, allowing for improved discrimination between specific pathogens and reducing false positives. Multimodal detection systems hold great potential in advancing food safety measures by offering robust, rapid, and accurate pathogen detection in a single integrated platform.

## 6. Conclusions

Ensuring the safety of food is of utmost importance, considering its significance to human existence and quality of life. Over time, pathogen detection in food has evolved from conventional methods such as culture-based techniques to more advanced approaches including PCR, ELISA, and antibody-based biosensors. The introduction of bioaffinity nanoprobes, particularly aptamers, has revolutionized biosensing due to their high sensitivity. The development of bioaffinity nanoprobes has also led to the emergence of technologies aimed at evaluating and analyzing their efficiency. Techniques such as SPR, FRET, and CD have proven valuable in assessing the performance of bioaffinity nanoprobes. Various biosensing technologies such as optical, colorimetric, and MIPs have emerged to detect foodborne pathogens and ensure food safety. Ongoing research and development in these fields aims to enhance their performance, sensitivity, and specificity, enabling more effective monitoring and ensuring the safety of our food supply. Advancements in biosensor technologies continue to play a crucial role in addressing the challenges associated with food safety and pathogen detection.

## Figures and Tables

**Figure 1 micromachines-14-01122-f001:**
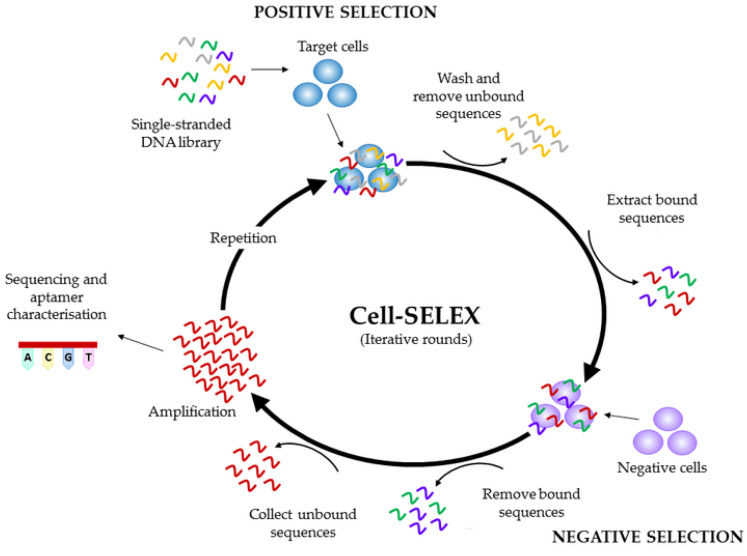
Schematic illustration of the SELEX process [40]. Reproduced with permission from Hays et al. (2017), ©MDPI, 2014 (Open access).

**Figure 2 micromachines-14-01122-f002:**
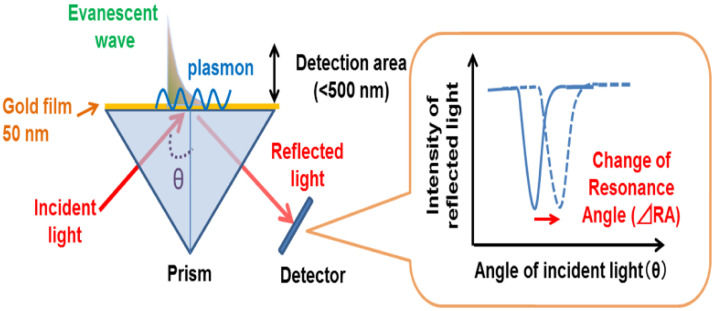
Schematic diagram of the working principle of surface plasmon resonance [58]. Reproduced with permission from Yanase et al. (2014), ©MDPI, 2014 (Open access).

**Figure 3 micromachines-14-01122-f003:**
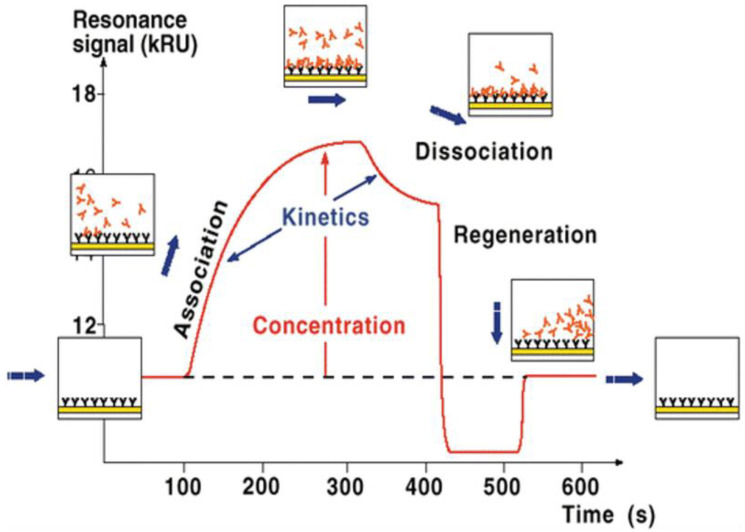
Sensorgram of an SPR analysis [65]. Reproduced with permission from Boutilier and Moulton (2017), © Springer, 2017.

**Figure 4 micromachines-14-01122-f004:**
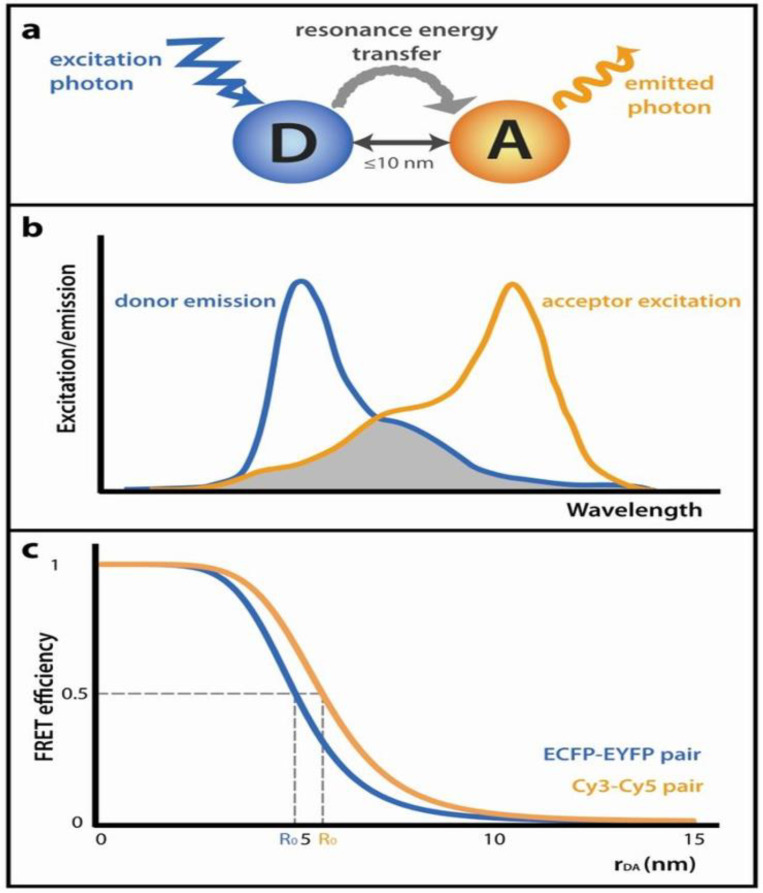
Förster resonance energy transfer (FRET) fundamentals: (**a**) A graphical illustration of FRET: An energized donor (D) employs a non-emissive method to relay its energy to an adjacent acceptor (A), prompting it to fluoresce. The gap between the fluorophores should be no more than 10 nm. (**b**) The emission peak of the donor must coincide with the excitation spectrum of the acceptor. The overlap region is indicated by the grey area. (**c**) The efficiency of FRET, represented as a function of the distance separating the donor and acceptor fluorophores (rDA) [81]. Reproduced with permission from Simkova et al. (2012), © MDPI, 2012 (Open access).

**Figure 5 micromachines-14-01122-f005:**
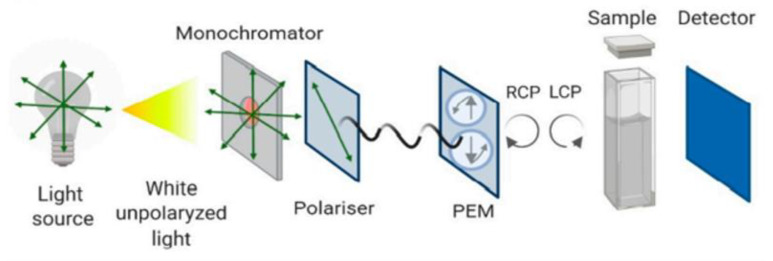
Schematic representation of the circular dichroism instrument configuration [89]. Reproduced with permission from Pignataro et al. (2020), © MDPI, 2020 (Open access).

**Figure 6 micromachines-14-01122-f006:**
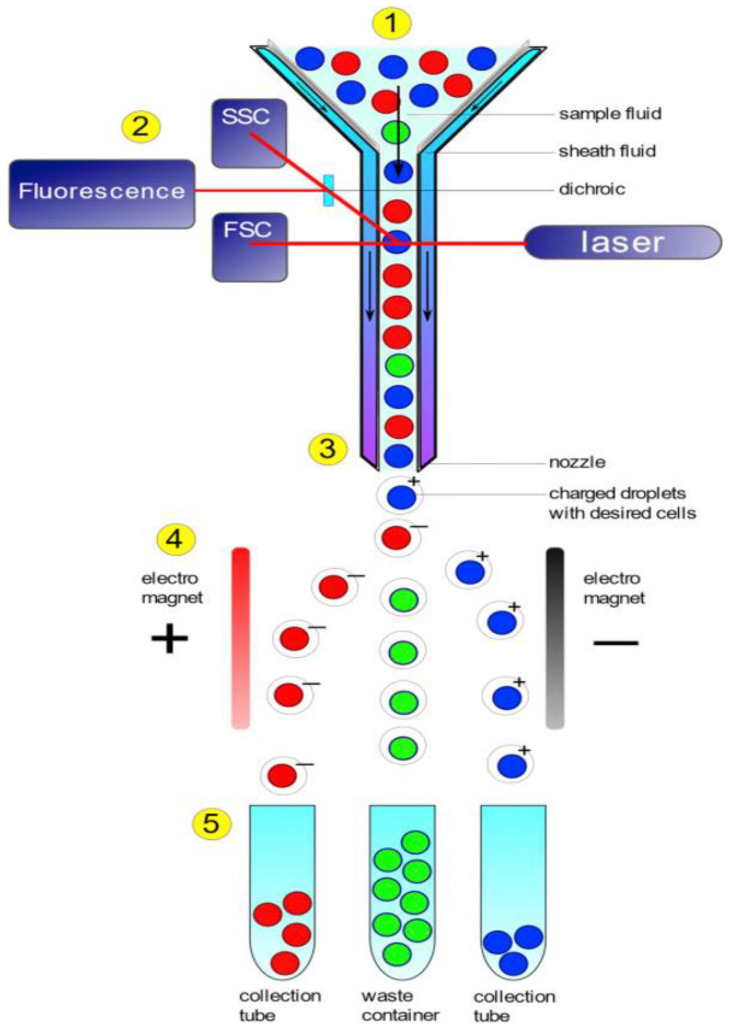
Components of a flow cytometer. Inside the flow cell (1), the fluid containing the cell sample is introduced at the center of the sheath fluid stream. To prevent mixing, these two fluids maintain a significant difference in velocity. This setup allows the cells to align in a single line, a process called hydrodynamic focusing. The aligned cells then pass through a laser and a series of detectors (2) that measure cell size using forward scatter (FSC), cell complexity using side scatter (SSC), and fluorescence. Before exiting the flow cell as individual droplets (3), the cells are selectively charged with electricity. Electromagnets (4) divert the droplets containing the targeted cells with a charge away from the main stream, guiding them into collection tubes positioned on the side (5). On the other hand, cells without a charge simply fall directly into a waste collection container [105]. Reproduced with permission from Bleichrodt and Read (2019), © Elsevier, 2019.

**Figure 7 micromachines-14-01122-f007:**
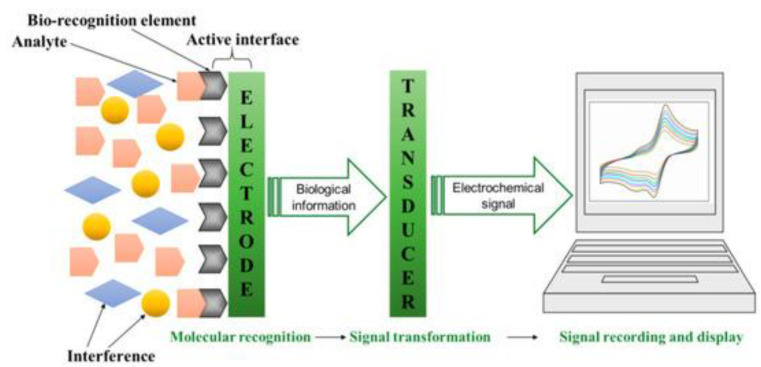
Schematic diagram of an electrochemical biosensor [106]. It consists of the sensing element (bio–recognition element), the transducer element, and the signal processor. The sensing element detects and binds to the target analyte, initiating a biochemical reaction. The transducer element converts this reaction into an electrochemical signal and the signal processor analyzes and interprets the signal, providing a measurable output. Reproduced with permission from Zhang et al. (2019), ©MDPI, 2014 (Open access).

**Figure 8 micromachines-14-01122-f008:**
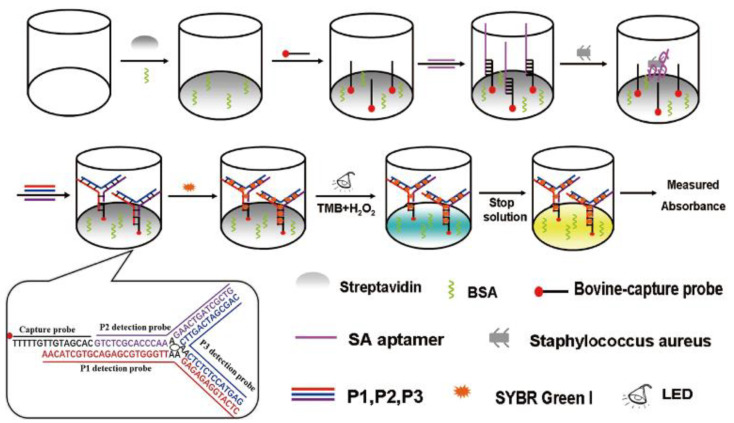
Schematic illustration of the staphylococcus aureus (SA) detection process involving a high-throughput colorimetric biosensor using aptamers and the photocatalytic activity of the dsDNA-SG I complex. This process begins with the coating of a 96-well plate with streptavidin. To prevent non-specific adsorption, bovine serum albumin (BSA) is used. Following this, a biotin-labeled capture probe is anchored to the plate surface via the streptavidin–biotin interaction. The SA-specific aptamer is then immobilized onto the 96-well plate via hybridization with the capture probe. When SA is present, the aptamer disengages from the capture probe–aptamer double strand due to a stronger interaction with SA. The resulting single-strand capture probe can hybridize with a DNA nanostructure, a three-way junction (TWJ), which consists of three detection probes (P1, P2, P3). Upon the addition of SG I, a dsDNA-SG I complex forms and catalyzes the oxidation of 3,3′,5,5′-tetramethylbenzidine (TMB) under LED photo-irradiation. The intensity of the resulting catalytic color is directly related to the number of bacteria present [122]. Reproduced with permission from Yu et al. (2020), ©Springer Nature, 2020 (Open access).

**Table 1 micromachines-14-01122-t001:** Comparison between monoclonal and polyclonal antibodies [19].

Category	Monoclonal Antibodies	Polyclonal Antibodies
Synthesis	Synthesized by one clone	Synthesized by numerous clones
	Production requires both in vitro and in vivo systems	Production is strictly in vivo (animal host is a must)
	Production requires trained personnel	Highly skilled personnel are not needed
	Short-term production is expensive but long-term production is cheap.	Short-term production is cheap but long-term production is expensive due to animal maintenance and deaths.
Homogeneity	They are homogenous in nature, making it easy to characterize their chemical nature and an easy choice for conjugation to different probes.	They are difficult to characterize since they are not homogenous.
Specificity	Highly specific	They are specific but exhibits cross reactivity
Degradation	Vulnerable to degradation under slightly harsh conditions.	Less vulnerable to degradation.
Affinity Purification	An excellent tool for affinity purification.	They are not a good choice for affinity purification

**Table 2 micromachines-14-01122-t002:** Summary of some examples of nanoprobe applications in foodborne pathogen detection.

Pathogen	Sample	Detection Method	Nanoprobe	Analysis Time (min)	LOD (CFU/mL)	Reference
*Salmonella typhymurium*	Milk	Electrochemical Impedance Spectroscopy (EIS)	Monoclonal antibodies	20	21	[139]
*Escherichia coli* O157:H7	Beef	Surface enhanced Raman spectroscopy	Aptamer	20	10^2^	[140]
*Vibrio parahaemolyticus*	Shrimp	Nuclear magnetic resonance spectroscopy	DNA	10	10^5^–10^8^	[141]
*Salmonella enterica*	Chicken	Differential pulse voltammetry	Aptamer	5	10	[142]
*Pseudocercospora fijiensis*	Banana	Surface plasmon resonance	Antibody	40	11.7 µg/mL	[143]
*Listeria*	Smoked salmon, milk, duck leg	Surface plasmon resonance	Antibody	60	10	[144]
*Campylobacter jejuni* *Staphylococcus aureus*	Chicken meat surface	Colorimetric	Antibody	120	10 100	[145]
*Escherichia coli*	Milk, water	Fluorescence	Nucleic acid	2.25	3.7 × 10^2^	[146]
Norovirus	Lettuce	Cyclic voltammetry	Concanavalin A	-	60 copies/mL	[147]
Diazinon	Chinese cabbage, tomato, apple	Fluorescence	DNA aptamers	-	-	[148]

## Data Availability

Not applicable.

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
