# Peer review of "Bioaffinity Nanoprobes for Foodborne Pathogen Sensing"

_micromachines, 2023, doi:10.3390/mi14061122_

Round 1

Reviewer 1 Report

I recommend that the manuscript be revised extensively before being considered for publication. The manuscript should provide an overview of the use of bioaffinity nanoprobes for foodborne pathogen sensing. There are a few areas that require attention and improvement, as outlined below:

1.      The authors have discussed the problem of foodborne illnesses caused by foodborne pathogens in the introduction and highlighted the limitations of conventional culture-based methods and other existing detection methods. The authors have then introduced nanosensors as a promising new technology for rapid and accurate detection of foodborne pathogens. However, the authors have not provided adequate background information on nanosensors or biosensor technology for readers who may not be familiar with the topic. Basics of nanosensors should be explained in the introduction and specific informations should be cited properly. For example in the first paragraph of the introduction no citations exist, however it contains specific information.

2.       In the “Overview of Bioaffinity Nanoprobes” section bioaffinity nanoprobes are given as antibodies, aptemers, enzymes, and non-antibody binding proteins. But no subtitle is given for non-antibody binding proteins. There are also no information about other synthetic nanoprobes like molecularly imprinted polymers which is another interesting topic and missing in the overall manuscript.

3.      In the “Analytical Approaches for Assessment of Bioaffinity Nanoprobes” section no information given for the electrochemical methods, however in the following section related with applications most of the examples include electrochemical methods.

4.      In the “Application of Bioaffinity Nanoprobes in Food Biosesnsing” section there are only two sub-titles including “electrochemical sensors” and “colorimetric sensors” which does not represent the whole literature. It would be helpful to provide more information on the specific probes that have been used in these sensors, as well as the sensitivity and specificity of each sensor. Additionally, the authors should consider adding information on other types of biosensors, such as optical and piezoelectric sensors. As the title of the manuscript indicates “bioaffinity nanoprobes”, it is better to classify the application studies according to bioreceptor used. As this part is the most important part of the review, more explanation should be given for the specific examples. Some of the studies are not explained at all. For example, Helali et al (105) and Shimaa et al (106). No bioreceptor indicated for these studies. Dong et al (104), abbreviation given for the material used. It should be explained first. There should be more comment about these studies, and for the other given examples within the manuscript.  

5.      The use of nanomaterials in the design of nanoprobes can also be discussed with a new heading.   

6.      All the manuscript and figures should be controlled again to avoid typing errors like “trancducer” in Fig7.

Overall, the manuscript provides an overview of the use of bioaffinity nanoprobes for foodborne pathogen sensing. However, there are a few areas that require improvement, as outlined above. With these revisions, the manuscript could be a valuable contribution to the field of food biosensing. I hope these comments are helpful to the authors and I look forward to seeing a revised version of the manuscript.

Author Response

  1. Some background information on nanosensors has been added to the introduction to give context to new readers. We have also included some citations to the first paragraph of the introduction.
  2. A new sub-section covering other bioaffinity nanoprobes, such as non-antibody binding proteins and molecularly imprinted polymers (MIPs), has been included.

  3. For the “analytical approaches for assessment of bioaffinity nanoprobes” section, the focus was discussing specific analytical tools and approaches for assessing the target binding properties of the bioaffinity nanoprobes, and not biosensing devices or applications utilizing the bioaffinity nanoprobes. However, in the section that follows, specific applications of the bioaffinity nanoprobes discussed and electrochemical biosensors were mostly used as the core approach since more advanced and have been demonstrated for various bioaffinity nanoprobes. It should be mentioned that the article is intended to focus more on the binding attributes/behaviors of the bioaffinity nanosensors for monitoring foodborne pathogens, and less about applications in biosensor devices/formats.

  4. Whilst the manuscript is intended to focus more on the binding attributes/behaviors of the bioaffinity nanosensors for monitoring foodborne pathogens, additional sections capturing other specific applications such as optical and piezoelectric sensor, making use of the bioaffinity sensors have been included to expand the scope on biosensor applications.

  5. The inclusion of a section on the use of nanomaterials in the design of nanoprobes would certainly expand the scope of the manuscript to cover the relationship between material properties and biosensor performance. However, this will be a derailment from the focus of the manuscript, opening up a whole new set of sections or topics that can be better communicated in another manuscript.

  6. The manuscript has been proofread to correct all typos and language errors.

The manuscript has been revised accordingly.

Reviewer 2 Report

Bioaffinity nanoprobes are used due to specific binding properties with target toxic food contaminants. In this review authors have discussed analytical techniques like SPR, Fluorescence Resonance Energy Transfer (FRET), circular dichroism, and flow cytometry for identitification and quantification of target nanoprobes designed for food sensing.

Some comments:

·      Please elaborate the figure caption. Describe the figure contents in detail, the contents of figure are too short to understand.

·      A table with examples of sensing methods, target, food materials, antibody/aptamer recognition would be good addition to the current manuscript.

·      Additionally, I suggest authors to following latest literature of toxic contaminants sensing from food, vegetables, fruits samples (by Hong group and others)--https://doi.org/10.31083/j.fbl2703092; https://doi.org/10.3390/ijms221910846; and https://doi.org/10.1186/s13765-023-00771-9.

·      Authors should give brief introduction to other kind of food sensors and summarize various contaminants.

English is readable.

Author Response

  1. The captions for the figures have been updated with more descriptive information.
  2. A new table has been added with information on sensing methods, target, food materials, antibody/aptamer recognition, limit of detection and specific references.

  3. The suggested references have been reviewed and cited in the revised manuscript.

  4. A new sub-section has been included to discuss other types of sensors for foodborne pathogen detection under the “application of bioaffinity nanoprobes in food biosensing” section. The sub-section also covers discussion on some specific contaminants detected using these technologies. Additional contaminants and detection methods have been included in the new Table 2.

Reviewer 3 Report

Titled Bioaffinity Nanoprobes for Foodborne Pathogen Sensing, this paper describes the types, advantages, disadvantages and applications of different nanoprobes. The writing of the article is not systematic and intuitive enough. Please refer to the following comments for major revisions.

1. Table 1 should add the descriptive information of the comparison. It is difficult for readers to understand which aspect of the comparison is made.

2. The author's article lacks a framework diagram for this article, and should make a framework diagram according to his own writing framework or understanding.

3. The pictures and text of the article are uneven, and even the text is deformed due to the stretching of the picture. The clarity of Figure 5 is poor. The pictures need to be reprocessed to meet the unification of the article format.

4. Line 464-480, why are there so many paragraphs?

5. In the fourth part (starting from line 431), a large number of documents in this part describe the current research status. These documents need to be summarized into a table, and information such as methods and nanomaterials should be indicated.

6. The research on microfluidics and smart phones that the author described in Part 5 is relatively mature and has a lot of research, which should not be included in this part. Since so many articles have already been studied, why doesn't the author discuss it in the main text but mention it in one sentence?

7. The author's full text is to introduce the advantages and disadvantages of the current nanoprobes with the feeling of popular science, but he does not describe the current research status in detail. The author's entire writing framework is not clear, and there is no detailed summary and induction. The sum table is not comprehensive enough, and a large number of pictures and documents of the research status should be added.

none

Author Response

  1. Another column has been added to Table 1 to provide more descriptive information of the comparison along with references.
  2. A framework diagram has been included in the revised manuscript.

  3. All the figures have been reprocessed and aligned properly.

  4. Some of the paragraphs in lines 464 – 480 have been collapsed to create fewer paragraphs.

  5. A new table has been included with information on sensing methods, target, food materials, antibody/aptamer recognition, limit of detection and specific references.

  6. A new section “Newer technologies” has been added to discuss advances in the application of microfluidics and smartphones to monitor foodborne pathogens.

  7. Additional information and figures have been included to cover the current state of research with respect to this topic.

Reviewer 4 Report

This is a review article on the study of the performance of bioaffinity nanoprobes. Currently, foodborne diseases are becoming increasingly serious, so a review of nanoprobes with bioaffinity for foodborne pathogens is of particular interest.  Hence, the topic dealt with in this paper is important

This paper is clearly structured and summarizes this field in terms of foodborne pathogen hazards and current status of research, types of bioaffinity probes and analytical methods, practical applications, and future prospects. The study focuses on a well-defined, meaningful, and relevant problem. The article could continue to be further revised to make it more readable. The following is a list of the main issues that the author needs to address.

1. The number of references cited in the article is quite large, but please try to cite some research results in the last five years to improve the advanced and representative nature of the article

2. The article is clearly structured and logically sound, and adequately introduces the performance analysis methods of the probe, but the practical application methods are listed in fewer categories, which can be appropriately increased to expand the breadth of the article.

3. The future outlook section presents its own thinking, but the details can still be considered to improve its summary and presentation.

4. In Table 1, please change this table to a three line table.

5. Some language and punctuation need to be reworked

none

Author Response

  1. More recent citations have been added to the manuscript.
  2. A new sub-section has been added under the “application of bioaffinity nanoprobes in food biosensing” section to expand the discussion on practical applications in biosensors.

  3. Further details and a cohesive summary have been included to improve the future outlook.

  4. Another column has been added to Table 1 to provide more descriptive information of the comparison along with references.

  5. The manuscript has been proofread to correct all typos and language errors.

Round 2

Reviewer 1 Report

Thank you for addressing the major corrections I raised in my previous review. I have carefully reviewed the revised manuscript, and I am pleased to see that you have made significant improvements in response to the comments. Your efforts have enhanced the quality and clarity of the paper.

I would like to commend you on your thoroughness in addressing the issues I identified. The revised manuscript now presents a more cohesive and well-structured argument. The modifications you have made have strengthened the overall flow of the paper, making it easier for readers to follow your line of reasoning.

Reviewer 3 Report

Agree to pubilish.

Reviewer 4 Report

No comments.